# Genetic and Epigenetic Pathogenesis of Acromegaly

**DOI:** 10.3390/cancers14163861

**Published:** 2022-08-10

**Authors:** Masaaki Yamamoto, Yutaka Takahashi

**Affiliations:** 1Division of Diabetes and Endocrinology, Kobe University Graduate School of Medicine, Kobe 650-0017, Japan; 2Department of Diabetes and Endocrinology, Nara Medical University, Kashihara 634-8521, Japan

**Keywords:** acromegaly, somatotroph tumors, *GNAS*, FIPA, miRNA

## Abstract

**Simple Summary:**

Various genetic and epigenetic factors are involved in the pathogenesis of somatotroph tumors. Although *GNAS* mutations are the most prevalent cause of somatotroph tumors, the cause of half of all pathogenesis occurrences remains unclarified. However, recent findings including the pangenomic analysis, such as genome, transcriptome, and methylome approaches, and histological characteristics of pituitary tumors, the involvement of AIP and GPR101, the mechanisms of genomic instability, and possible involvement of miRNAs have gradually unveiled the whole landscape of underlying mechanisms of somatotroph tumors. In this review, we will focus on the recent advances in the pathogenesis of somatotroph tumors.

**Abstract:**

Acromegaly is caused by excessive secretion of GH and IGF-I mostly from somatotroph tumors. Various genetic and epigenetic factors are involved in the pathogenesis of somatotroph tumors. While somatic mutations of *GNAS* are the most prevalent cause of somatotroph tumors, germline mutations in various genes (*AIP*, *PRKAR1A*, *GPR101*, *GNAS*, *MEN1*, *CDKN1B*, *SDHx*, *MAX*) are also known as the cause of somatotroph tumors. Moreover, recent findings based on multiple perspectives of the pangenomic approach including genome, transcriptome, and methylome analyses, histological characterization, genomic instability, and possible involvement of miRNAs have gradually unveiled the whole landscape of the underlying mechanisms of somatotroph tumors. In this review, we will focus on the recent advances in genetic and epigenetic pathogenesis of somatotroph tumors.

## 1. Background

Excessive secretion of GH and IGF-I prior to closure of the epiphyseal line results in giantism with marked increases in height, while increased secretion of these hormones post-closure results in acromegaly. Both gigantism and acromegaly are relatively rare, occurring in only 8 and 11 cases per million people per year, respectively [1]. Most of these cases result in somatotroph tumors except for extremely rare causes such as orthotopic/ectopic GHRH-producing tumors and ectopic GH-producing tumors [2,3]. The 2022 WHO classification defined somatotroph tumors originating from PIT-1 lineage PitNETs. Although they are of PIT-1 lineage, PitNETs are classified into nine types. Acromegaly is mainly caused by somatotroph tumors such as densely granulated somatotroph tumors and sparsely granulated somatotroph tumors, while a minority of cases result from mammosomatotroph tumors and mature plurihormonal PIT1-lineage tumors [4].

Densely granulated somatotroph tumors are usually diagnosed at a younger age and when they are smaller than sparsely granulated somatotroph tumors due to being highly hormonally active [4]. Densely granulated somatotroph tumors tend to exhibit low signal intensity on T2-weighted imaging and a good response to somatostatin analogs (SSA), while sparsely granulated somatotroph tumors tend to be characterized by larger and invasive tumors with high signal intensity on T2-weighted imaging, which are resistant to SSA [5,6].

The genetic abnormalities that cause somatotroph tumors can be broadly classified into germline and somatic mutations, but recent advances suggest that these abnormalities are not only gene mutations but also copy number variations and epigenetic abnormalities. Although 40% of somatotroph tumors are caused by somatic mutations in the *Guanine nucleotide activating subunit* (*GNAS*) gene and the additional portion can be explained by the handful of classified rare germline mutations [7], the cause of almost half of all somatotroph tumors remains unclarified. 

## 2. Genetic Pathogenesis in Sporadic Somatotroph Tumors

Pituitary adenomas occur sporadically in more than 95% of cases, with most of the genetic causes of these conditions are associated with somatic, mosaic, and familial hypoosmotic mutations, all of which are monoclonal [8,9,10]. Despite this knowledge, our understanding of these conditions remained reasonably limited until the advent of whole genome sequencing, which has catalyzed significant advances in our understanding of the pathophysiology of sporadic pituitary adenomas [11].

A recent report of a whole genome analysis comprising 771 patients with pituitary adenomas and 2788 healthy controls from China revealed significant susceptibility loci at chromosomes 10p12.31, 10q21.1, and 13q12.13, but showed no specific genetic mutations associated with this condition [12]. Whole-exome sequencing studies of pituitary tumors showed that the number of somatic mutations in the coding regions was not high (less than 10 per tumor sample) [13,14,15,16,17,18].

### Somatic GNAS Mutations in Somatotroph Tumors

The most common cause of somatotroph tumors is somatic mutations in the *GNAS* gene, 20q13.3 [19,20]. These mutations result in amino acid substitutions at Arg201 and Gln227, resulting in the constitutive active mutation of the Gsα subunit [20]. These lead to excessive cAMP and PKA signaling, resulting in autonomous secretion of GH [20]. The frequency of these mutations in sporadic cases for adults with somatotroph tumors varied between studies, which widely ranged from 4.4% to 53% by study design, sample size, race, country, and method for DNA sequence [21,22,23,24,25,26]. Whole genome and exome sequencing studies of somatotroph tumors have also shown that *GNAS* mutation rates can reach up to 25% and 31%, respectively [27,28]. Patients with somatic *GNAS* mutations tend to be relatively elderly at diagnosis and present with smaller, less invasive tumors. In general, most studies suggest that *GNAS* mutations are associated with densely granulated somatotroph tumors, and the efficacy of first-generation SSA is higher [29]. However, controversy remains with regards to the association between *GNAS* mutations and pathological subtype, and a small number of cases exhibit the atypical phenotype including young age, macroadenoma, invasiveness and resistance to SSA and dopamine agonist (DA) [30,31,32].

## 3. Germline Mutations in Somatotroph Tumors

A small proportion of somatotroph tumors are caused by germ cell mutations in various genes including *MEN1*, *CDKN1B*, *PRKAR1A*, *GNAS*, *AIP*, *GPR101*, *SDHx*, *MAX, NF1, IGFS1* and *TSC* (Table 1). In these cases, they are generally familial and observed in young patients.

### 3.1. Familial and Inherited Syndromes

Familial onset and genetic tumor syndromes are found in approximately 5% of all pituitary tumors, and the causative gene has been identified in some cases. In general, pituitary adenomas caused by genetic mutations are characterized by a younger age of onset, accelerated tumor growth and invasion, and resistance to surgery and medical therapy, especially to first-generation somatostatin analogs, making them more clinically challenging [10].

### 3.2. Multiple Endocrine Neoplasia Type 1 (MEN1)

*MEN1*, which encodes the tumor suppressor protein Menin [33], is known to be one of the most critical genetic targets in somatotroph tumors with more than 1300 germline mutations recorded in this locus. While these mutations occur throughout the coding sequence, it is interesting to note that the majority of the inactivating mutations occur at the splice sites resulting in distinct frameshifts [34]. MEN1 has an autosomal manifest mode of inheritance, with most neoplastic lesions described in the pituitary, parathyroid, and pancreas, although these neoplasms can occur at other sites as well [35]. A recent report describing the 560 Japanese patients registered in the MEN Consortium revealed that the mean age at diagnosis was 48 years, with 94% presenting with parathyroid tumors, 59% with pancreatic endocrine tumors, 50% with pituitary tumors, 20% with adrenocortical tumors, and 8% with foregut carcinoid tumors. This report also noted a frequency of complications at 98% penetration by the age of 50 years [36]. In addition, 17% of the adult patients and 30% of the child patients with *MEN1* mutations presented with pituitary adenomas as their initial manifestation, with most of these events being recorded between their childhood and 40 years of age, although the cohort presented with a wide age range (from 5 to 90 years) for initial onset. Lactotroph tumors were the most frequent pituitary tumors, followed by nonfunctioning adenomas, with somatotroph tumors accounting for 10% of them [37]. Characteristics of somatotroph tumors in *MEN1* mutations include local invasion, multiple hormone-producing potential, and often treatment-resistant macroadenomas.

Menin, a member of the histone methyltransferase complex, is known to be involved in the regulation of *CDK*, *CDKN1B*, and *CDKN2C* expression and thus of the cell cycle. Therefore, it is considered that tumors associated with *MENIN* mutations are the result of cell cycle abnormalities, although the detailed mechanism remains unclear [38,39]. Although the *MEN1* genotype–phenotype correlation is not clear, it is recommended that *MEN1* mutation carriers undergo periodic imaging and endocrinological testing to facilitate early cancer detection, especially for thymic carcinoids [35]. Genetic testing guidelines recommend that *MEN1* mutation carriers begin routine imaging and endocrinological screening from 5 years of age [35].

### 3.3. Multiple Endocrine Neoplasia Type 4 (MEN4)

*CDKN1B* genes with loss-of-function mutations have been identified in 10–20% of cases with a similar phenotype to that of MEN1 but without any *MENIN* mutations, and they are referred to as *MEN4* mutations. Somatotroph tumors and corticotroph tumors are the most common [40,41], but these patients often present with other complications including carcinoids, reproductive system tumors, cervical neuroendocrine tumors, and adrenal and renal tumors.

### 3.4. Carney Complex (CNC)

CNC is clinically diagnosed when two or more of the following are present: skin pigmentation, cardiac myxomas, primary pigmented nodular adrenocortical lesions (PPND), large cell calcifying Sertoli cell tumors, ductal adenomas, pustular melanomas, blue nevus, osteochondral myxomas, thyroid tumors, and acromegaly. This disease is inherited as an autosomal manifestation, with de novo mutations in approximately 30% of cases. The average age at diagnosis is normally over 20 years of age, and most cases are caused by inactivating mutations in the *PRKAR1A* gene (CNC1) at 17q24.2. However, 20% of CNC cases are caused by abnormalities in CNC2 located at 2p16 (causative gene unidentified). The incidence of acromegaly is similar between CNC1 and CNC2 [42], and several other cases have referenced additional abnormalities in *PRKACB* [43]. As many as 75% of CNCs present with elevated GH/IGF-I and PRL levels, and 10–12% of these cases including somatotroph tumors are slightly more common in females, developing one to two decades earlier than sporadic cases [44]. Pathologically, these somatotroph tumors can be described as either adenoma or hyperplasia, with both commonly presenting as multiple foci with a mixture of GH- and PRL-producing cells. Treatment is the same as for common somatotroph tumors, although some cases have been reported to be resistant to SSA treatment [45].

### 3.5. McCune–Albright Syndrome (MAS)

McCune–Albright syndrome, first described in 1937, is a term used to describe a triad of symptoms including fibrous osteodysplasia, precocious puberty, and café-au-lait spots. This condition is often complicated by endocrine abnormalities including gigantism/acromegaly, thyroid gland abnormalities, nodular adrenal hyperplasia and hyperparathyroidism, is secondarily caused by vitamin D deficiency unlike the primary cause, and has an estimated prevalence of 1/100,000 to 1/1,000,000 [46]. Somatotroph tumors in MAS are caused by somatic mosaic gain-of-function mutations in the *GNAS1* gene [46], and 75% of these somatotroph tumors are found in male MAS patients, with the majority of these cases occurring at approximately 20 years of age. That being said, there are many instances of somatotroph tumors in children with MAS as well [47,48]. In addition, 71–92% of somatotroph tumor cases exhibited GH- and prolactin-producing tumors [49]. The overproduction of GH and IGF-I induces increases in the common fibrous dysplastic bone deformities of the skull associated with MAS [48,49]. These deformities can impact the optic or auditory nerves by compression [50,51]. In addition, detection of the underlying genetic mutations can be difficult, as these patients present with a DNA mosaic that often results in mixtures of DNA profiles in the peripheral blood lymphocytes, and thus, these mutations must be confirmed using tissue DNA from the lesion site. Recent reports suggest that digital PCR using whole blood or cell-free DNA can be used to identify mosaicism with up to 80% sensitivity [52]. Treatment in these patients is also often complex as surgical removal is not always possible in cases of concomitant skull base fibrous osteodystrophy because of the complex nature of these bone deformities and the abundance of nutrient vessels in these lesions. In these cases, some patients can be treated with SSA or radiation therapy, but this is usually a last resort owing to concerns around radiation-induced sarcomatoid changes [49].

### 3.6. Familial Isolated Pituitary Adenoma (FIPA)

FIPA is defined as the presence of two or more related members exhibiting pituitary adenomas without known genetic causes. Although the prevalence of this condition is rare, more than 500 families have been identified so far [53]. Thirty-five percent of FIPA patients present with somatotroph tumors, with most pituitary adenomas developing half a decade to two decades earlier than cases with sporadic pituitary adenomas [54]. These patients also often present with relatively large tumors with a more invasive phenotype [55]. Mutations in *AIP* and *GPR101* that cause X-LAG syndrome and chromosome Xq26.3 microduplication comprise approximately 20% of FIPA cases; however, the sporadic occurrence of FIPA in patients without a family history of pituitary tumor is rare [56,57].

### 3.7. Aryl Hydrocarbon Receptor-Interacting Protein (AIP)

*AIP*, located at 11q13.2, was identified as the causative gene in a case of familial GH-producing adenoma in a family from northern Finland in 2006. *AIP* mutations are found in 15–20% of FIPA cases and 40–50% of familial GH-producing adenomas, with a penetrance of 13–30%. In general, the age of onset in *AIP* mutation-positive cases is less than 30 years, with 65% of cases occurring before the age of 18 [58]. These cases often present with large and aggressive tumors, especially in childhood-onset cases, which are prone to infiltration and extension beyond the sella turcica and cause pituitary apoplexy [42]. In contrast, the frequency of somatic mutations in *AIP* in sporadic cases of pituitary adenoma is 4%, and there are no reports of germline mutations in sporadic cases so far [59]. As an underlying mechanism, AIP is a co-chaperone protein linked with PDE4A5, a cAMP-degrading enzyme, and *AIP* mutations are thought to reduce PDE4A5 inhibition and increase intracellular cAMP levels, leading to GH-autonomous secretion and pituitary tumorigenesis [60].

### 3.8. X-Linked Acrogigantism (X-LAG)

X-LAG is characterized by the microduplication of the Xq26.3 region in embryonic or somatic cells, which results in the overexpression of the *GRP101* gene. Overexpression of the GRP101 protein induces the constitutive activation of G proteins such as Gs and Gq/11 and stimulates GH release by overproduction of cAMP via protein kinase A and protein kinase C [61,62]. X-LAG has been reported in patients younger than 40 years of age, with onset typically seen as early as 1 year of age. In addition, nearly all X-LAG patients are diagnosed with GH- or PRL-producing pituitary tumors by 3 years of age [61,63]. More than 85% of patients present with mixed GH and PRL-producing tumors, and pathology varies from hyperplasia to macroadenoma, but the MIB-1 index is often low [64]. These patients usually present with normal height and weight at birth but demonstrate marked growth acceleration during the first 2 years of life. Their phenotype is similar to the general phenotype of gigantism and acromegaly, except that 30% of patients have increased appetite and fasting hyperinsulinemia, while another 20% of patients present with melanoderma. Cure by tumor resection alone is often difficult due to the large size of the tumor, and additional drug and radiation therapy are often required [63,65].

### 3.9. Succinate Dehydrogenase (SDHx)

Loss-of-function mutations in the succinate dehydrogenase (SDH) gene cause pheochromocytoma/paraganglioma (PPGL), which has been associated with the development of pituitary adenomas, even though the penetration of this complication is low. Only four cases of adenoma development have been reported so far, with patients ranging in age from 37 to 84 years. These pituitary adenomas tend to be aggressive and have large tumor sizes, making this uncommon complication significant in a clinical context [66].

### 3.10. MYC-Associated Factor X (MAX)

Germ cell mutations in *MAX* located at 14q23.3 are associated with neuroendocrine and renal tumors, as well as small-cell lung cancer and both somatotroph tumors and PRLomas. The reported three cases have all required drug and radiation therapy in addition to surgical intervention to facilitate clinical treatment, likely owing to the large tumor size in these young patients [67].

### 3.11. Neurofibromatosis Type 1 (NF1)

The *NF1* gene encodes neurofibromin, a GTPase-activating protein that suppresses downstream signaling pathways such as MAPK and PI3K by inactivating RAS. Patients with NF1 present with very diverse clinical conditions, including cutaneous neurofibromas, café-au-lait spots, freckles in the groin and axillae, iris nodules, and optic chiasm gliomas, two or more of which are diagnostic for NF1.

GH excess has been observed in cases with *NF1* mutations. While the underlying mechanism as to how *NF1* mutations induce GH excess remains unknown, some hypotheses have been proposed. The first hypothesis proposes that somatostatin neurons in the hypothalamus may be disrupted by infiltrating optic glioma [68]. Another is that undetermined genetic or epigenetic mechanisms may upregulate GPR101 expression, which results in GH excess/ somatotroph tumors [68]. Several cases of NF1-associated acromegaly caused by somatotroph tumors have been reported [68,69].

### 3.12. Tuberous Sclerosis Complex (TSC)

Tuberous sclerosis is an autosomal dominant disease characterized by constitutive activation of mTORC1 signaling caused by loss-of-function mutations in either *TSC1* at 9q34.13, or *TSC2* at 16p13.3. Phenotypes of this disease include malignant tumors of the brain, lungs, heart, skin, and kidneys, as well as epilepsy, intellectual disability, and autism. To date, of the four *TSC* mutation-derived pituitary tumor cases in the literature, only one has included the development of somatotroph tumors [70].

## 4. Chromosomal Alterations and Pituitary Tumorigenesis

Whole genome/exome sequencing was used to examine chromosomal alterations in pituitary adenomas and frequently identified either chromosome loss or gain for chromosomes 1, 2, 7, 8, 11, 18, 19, and 22 [71]. In addition, other research has shown that *Pituitary Tumor-transforming Gene 1 (PTTG1)* overexpression in pituitary adenomas leads to chromosomal instability and aneuploidy [72], with a recent CGH analysis reporting that aneuploidy is more frequently observed in invasive adenomas [73]. In addition, whole exome sequencing detected arm-level somatic copy number alterations (SCNA) in 42 samples from pituitary macroadenomas at extensive sites across the genomes in 29% of specimens. Chromosomal alterations were shown to be more frequent in hormone-producing adenomas, particularly somatotroph tumors and null-cell adenomas [15,28,71]. In contrast, copy number alterations were less frequent in nonfunctioning adenomas and gonadotrophin-producing adenomas [14,74]. Another study of 159 pituitary adenomas revealed that SCNA was critical to the production of hormone-producing adenomas, with far less frequent chromosomal aberrations in nonfunctioning adenomas. Evaluations of the single-gene SCNA pathway revealed a significant role for the cAMP pathway in somatotroph tumors, with both GH production and DNA damage closely linked to changes in cAMP activation by GHRH analogs. These studies also showed that GH hypersecretion was associated with SCNA and genomic instability [15]. DNA replication stress, cell proliferation, and the role of homeostatically elevated cAMP resulting in over-secretion of GH may predispose senescence instead of apoptosis to pituitary cells [75]. Analysis of DNA methylation profiles showed that increased expression of the GH and SST5 genes in somatotroph tumors is associated with decreasing methylation of their respective promoter regions [13]. DNA hypomethylation and higher mRNA expression of KCNAB2 were observed in somatotroph tumors compared to non-functioning pituitary adenoma, which may be involved in the over-secretion of GH and its tumorigenesis [76,77].

## 5. Non-Coding RNA in Somatotroph Tumors

MicroRNAs (miRNAs) are 21–25 nucleotide-length single-stranded RNA molecules that are involved in the post-transcriptional regulation of gene expression in eukaryotes. miRNAs bind to their target mRNAs with incomplete homology and generally recognize the 3′UTR of the target gene, thereby destabilizing the target mRNA and suppressing protein production through translation repression. miRNA-mediated transcriptional repression is known to play important roles in various pathways in tumors [78]. Therefore, the pathophysiological relevance of miRNAs in somatotroph tumors reported to date is summarized as follows (Table 2).

miR-15a and miR-16-1 were downregulated in somatotroph tumors [79], and miR-16 reportedly targets *GHR*, *IGF-1*, *IGF1R*, and *IGF2R* expression [80]. Downregulated miR-34b, miR-34b, miR-326, miR-432, miR-548c-3p, miR-570, and miR-603 expression was associated with increased *HMGA1*, *HMGA2*, and *E2F1* expression in somatotroph tumors [81]. miR-128, which directs B-lymphoma Mo-MLV insertion region 1 (BMI1), was reduced in somatotroph tumors via regulating the PTEN-AKT pathway [82]. In another report, miR-23b and miR-130b were significantly reduced in pituitary adenomas including somatotroph tumors. Moreover, the expression level of *HMGA2* and *cyclin A2 (CCNA2)* that these miRNAs target was upregulated in human pituitary adenomas [83]. Downregulated miR-185 expression was a predictive marker of the effect of SSA in patients with acromegaly [84]. An upregulated expression of miR-338-3p was observed in invasive somatotroph tumors, concomitant with the increased expression of *PTTG* [85]. miR-423-5p, which inhibits PTTG1 expression, exhibited decreased expression in somatotroph tumors [86]. miR-107 was upregulated in sporadic somatotroph tumors, which binds to 3′UTR and suppresses *AIP* expression [87]. miR-26b promotes PI3K/AKT pathway via inhibiting PTEN [82]. miR-184 was significantly upregulated in somatotroph tumors, although its pathophysiological significance remains unclear [88]. miR-21-5p was highly expressed in somatotroph tumors compared to non-functioning pituitary adenomas [89]. It was abundantly contained in exosomes derived from somatotroph tumors and regulated the PDCD4/AP-1 pathway by targeting PDCD4 and SMAD Family Member 7 (Smad7) [89].

Long noncoding RNAs (lncRNAs) are a form of noncoding RNA transcript longer than 200 bases that have a wide range of functions, such as chromatin modification, transcriptional regulation, and post-transcriptional regulation [90]. In pituitary adenomas including somatotroph tumors, lncRNA H19 exhibited reduced expression compared with that of the normal pituitary gland, and it presented a negative correlation with tumor volume via blocking mTORC1-mediated 4E-BP1 phosphorylation [91]. Another group reported that H19 expression in invasive somatotroph tumors was significantly higher than that in non-invasive somatotroph tumors; however, the underlying mechanism remains unknown [92].

While the lncRNA MEG3 is known to inhibit cell proliferation [93], two independent studies have reported the involvement of the lncRNA MEG3 in the development of somatotroph tumors [94,95]. Furthermore, MEG3 expression levels in somatotroph tumors were positively correlated with patient serum GH and IGF1 levels and conversely negatively correlated with tumor size. More interestingly, MEG3 expression was significantly increased and tumor proliferative and invasive potential was decreased in comparison between the *GNAS* mutant and wild-type groups (Table 2) [95].

## 6. Conclusions

Advances in genome analysis methods have led to the identification of several additional novel causative genes to *GNAS* mutation over the last 20 years. These evaluations have also facilitated the evaluation of various pathological mechanisms in these tumors, revealing that copy number variations and epigenetic abnormalities are also critical components in somatotroph tumors development. However, the cause of the majority of cases with somatotroph tumors remains unelucidated, highlighting the need for additional breakthrough studies. Recent advances in therapeutic strategies have enabled “a targeted therapy” based on the underlying mechanisms and responsible molecules. Therefore, it is critical to clarify the remaining causes to develop tailor-made therapies.

## Figures and Tables

**Table 1 cancers-14-03861-t001:** The list of disease-related genes that are involved in the pathogenesis of somatotroph tumors. ND: not determined.

Germline or Mosaic Mutations	Reported Incidence of Mutations	Concomitant Development of the Tumors	The Mechanisms of Tumorigenesis by the Gene Mutation
*MENIN* (*MEN1*)	1/30,000~1/40,000	parathyroid adenoma, pancreatic neuroendocrine tumor	Influence on cell proliferation, cell signaling, transcriptional regulation, and genome stability
*Cyclin Dependent Kinase Inhibitor 1B * (*CDKN1B*)	up to 3% of cases with negative *MEN1* mutation	corticotroph adenoma, parathyroid adenoma	Dysregulation of the cell cycle.
*Protein Kinase CAMP-Dependent Type I Regulatory Subunit Alpha* (*PRKAR1A*)	750 cases	skin lesions, cutaneous and heart myxomas, PPNAD, large cell calcifying Sertoli cell tumor/ calcification of testis, thyroid carcinoma or multiple hypoechoic nodules, breast ductal adenoma psammomatous melanotic schwannomas, blue nevus, osteochondromyxoma	Inactivating mutations of PRKAR1A lead to uncontrolled activation of cAMP-dependent kinase activity in affected tissues
*GNAS Complex Locus* (*GNAS1*)	1/100,000~1/1,000,000 live births	fibrous dysplasia, precocious puberty, café-au-lait skin lesions	A constitutively activated cAMP pathway leading to persistent GH hypersecretion and cell proliferation.
*Aryl Hydrocarbon Receptor-Interacting Protein* (*AIP*)	10% of FIPA	none	Elevated concentrations of cAMP
*G Protein-Coupled Receptor 101* (*GPR101*)	7.8–10% of gigantism patients	none	Activation of an orphan G protein-coupled receptor and increased cAMP levels, which is a key factor in GH secretion and cell proliferation in response to GHRH
*Succinate dehydrogenase* (*SDHx*)	very rare	pheochromocytoma/paraganglioma (PPGL)	The accumulation of onco-metabolites that inhibit degradation of hypoxia transcription factor (HIFα)
*MYC associated factor X* (*MAX*)	very rare	pheochromocytoma/paraganglioma (PPGL), neuroendocrine cells, renal tumors, small cell lung cancer	To interact with other parts of the MAX-MLX network, which is responsible for the integration of cellular signals and modulates the expression of another gene
*Neurofibromatosis type 1* (*NF1*)	1:2500–1:3500 live births.	optic pathway gliomas, cutaneous neurofibromas, cafe-au-lait skin lesions, intertriginous freckling, Lisch nodules, brain tumors	Involved in cell growth and proliferation, by inhibiting RAS activity and regulation of cAMP levels
*Tuberous sclerosis complex* (*TSC*)	very rare	multiple hamartomas in brain, lungs, heart, skin, and kidney	ND

**Table 2 cancers-14-03861-t002:** The list of miRNAs and lncRNAs and their target mRNAs involved in the pathogenesis and characteristics of somatotroph tumors.

**miRNA**	**Expression**	**Target Genes (Putative)**
miR-15a, miR-16-1	down regulated	*GHR*, *IGF-1*, *IGF1R*, *IGF2R*
miR-34b, miR-326, miR-432, miR-548c-3p, miR-570, miR-603	down regulated	*HMGA1*, *HMGA2*, *E2F1,*
miR-128	down regulated	*BMI1*
miR-23b,	down regulated	*HMGA2*
miR-130b	down regulated	*CCNA2*
miR-185	up/down regulated	*SSTR2*
miR-338-3p, miR-423-5p	up regulated	*Pttg1*
miR-107	up regulated	*AIP*
miR-26b	up regulated	*PTEN*
miR-184	up regulated	*IGF1R*
miR-21-5p	up regulated	*PDCD4* and S*mad7*
**lncRNA**	**Expression**	**Target**
H19	down regulated	*4E-BP1*
	up regulated	not determined
MEG3	up regulated	not determined

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
