# Peer review of "Genetic and Epigenetic Pathogenesis of Acromegaly"

_cancers, 2022, doi:10.3390/cancers14163861_

Round 1

Reviewer 1 Report

1, It is suggested that the latest WHO classification of GH adenomas can be described in the background. Meanwhile, the characteristics and clinical differences of densely/sparsely
 granulated somatotroph tumour are briefly described.

2, Line 43, “Genetic pathogenesis in sporadic pituitary adenoma” might be “Genetic pathogenesis in sporadic GHoma”?

3, Line 56,if there is not the 2.2 subheading, 2.1 subheading should be deleted.

4, in the section of “2.1 Somatic GNAS mutations in GHoma”, GNAS mutations have pointed out the association with densely granulated subtype, but several subsequent articles have pointed out that the mutations in GNAS are not associated with pathological classification, and do not indicate the prognosis and the response to SSA drugs (European Journal of Endocrinology, 2013;168(4):491-9. )

5, line 68, in this section of “Germline mutations in GHoma”, the author should provide a table, in which each gene mutations is explained in detail, including the incidence of mutations, concomitant development of the tumors, and the clinical significance of the gene mutation. This makes the reader very clear.

6. in the section of “Germline mutations in GHoma”, KCNAB2 is also associated with tumorigenesis and secretion of acromegaly (J Neurosurg,2020,28;134(3): 787-793).

Reviewer 2 Report

Dear Authors,

This is an interesting review.

Here are my observations/suggestions/comments:

1.    Abstract. In endocrinology, the term of “GH-oma” is rarely use, rather somatotropinoma

2.    Introduction - “Both of these conditions” This is the same condition, the patient’ s age at onset is different.

3.    Introduction - “except for extremely rare causes such as ectopic GHRH-producing tumors and ectopic GH-producing tumors” – and also hypothalamic GH-RH producing tumors, not only ectopic GH-RH excess

4.    Introduction - Lines 33-35. There is no connection between the clinical appearance, meaning the term “acromegaly” and the familial inheritance. Nowadays, there are genetic forms with early detection without any acral changes.

5.    “The most common cause of GHoma is somatic mutations in the GNAS gene, 20q13.3, 57 accounting for 30-50% of sporadic GHomas” – You should clearly specify in what population or study because this is not generally applicable.

6.    Table 1 mostly introduces germline mutations; the statement (line 57) involves somatic mutations. Please re-structure it.

7.    Line 68-78. Another specific feature of this type is the presence of a large tumor. Also, the resistance to the other lines of medical therapy is presented, not only to first generation of somatostatin analogues. Please restructure the paragraph.

8.    Line 80 - “tumor suppressor Menin”, do you mean tumor suppressor protein Menin?

9.    Line 101 - “MEN1 tumors” There is no MEN1 tumors since “MEN” means multiple endocrine neoplasia. Please revise the term

10. Line 109 - “have been identified in 10-20% of all cases presenting” – Which “all cases”?

11. Line 131 - “SSA” – please explain abbreviations when first used

12. Line 136 – primary hyperparathyroidism – Please specify the type of hyperparathyroidism since not all of them are typically a part of the syndrome

13. Line 205 - “Incidence of these tumors is low” What do you mean by “low” (because the entire syndrome is exceptionally rare)?

14. Line 214 -“GH excess in NF1 is not usually the result of a GHoma but” – This is not clear

15. Line 219 - “autosomal manifestation of an inherited disease” – do you mean autosomal dominant?

16. Conclusion - “However, the cause of approximately 50% of GHoma remains unclarified, highlighting the need for additional breakthrough studies.”  The percent is much higher in daily practice. I do not agree with this data.

Best regards,

Thank you,

Round 2

Reviewer 1 Report

none.

Reviewer 2 Report

Dear Authors,

I agree with the current form of the article and recommend its publication.

Thank you